# Advances in Bacteriophage Therapy against Relevant MultiDrug-Resistant Pathogens

**DOI:** 10.3390/antibiotics10060672

**Published:** 2021-06-04

**Authors:** Antonio Broncano-Lavado, Guillermo Santamaría-Corral, Jaime Esteban, Meritxell García-Quintanilla

**Affiliations:** Department of Clinical Microbiology, IIS-Fundación Jiménez Díaz, Av. Reyes Católicos, 2, 28040 Madrid, Spain; antonio.broncano@fjd.es (A.B.-L.); guillermo.santamaria@quironsalud.es (G.S.-C.); jesteban@fjd.es (J.E.)

**Keywords:** bacteriophage, alternative therapy, phage therapy, multiresistant, *Acinetobacter baumannii*, *Klebsiella pneumoniae*, *Escherichia coli*, *Pseudomonas aeruginosa*

## Abstract

The increase of multiresistance in bacteria and the shortage of new antibiotics in the market is becoming a major public health concern. The World Health Organization (WHO) has declared critical priority to develop new antimicrobials against three types of bacteria: carbapenem-resistant *A. baumannii*, carbapenem-resistant *P. aeruginosa* and carbapenem-resistant and ESBL-producing *Enterobacteriaceae*. Phage therapy is a promising alternative therapy with renewed research in Western countries. This field includes studies in vitro, in vivo, clinical trials and clinical cases of patients receiving phages as the last resource after failure of standard treatments due to multidrug resistance. Importantly, this alternative treatment has been shown to be more effective when administered in combination with antibiotics, including infections with biofilm formation. This review summarizes the most recent studies of this strategy in animal models, case reports and clinical trials to deal with infections caused by resistant *A. baumannii*, *K. pneumoniae*, *E. coli,* and *P. aeruginosa* strains, as well as discusses the main limitations of phage therapy.

## 1. Introduction

The phenomenon of antimicrobial resistance is a paradigm of evolution. Once a new antibiotic is discovered, it is followed by the detection of resistance among those bacteria that are usually susceptible to it. The sum of different resistance mechanisms leads to the development of resistant phenotypes among bacteria. These pathogens are the cause of many human infections, which can become extremely difficult to treat. Nowadays, the increase of multiresistance in bacteria is becoming a major public health problem, especially in hospital settings with increased dissemination of resistant strains due to the globalized world [1]. The most important bacteria in this regard are carbapenem-resistant *Enterobacteriaceae* (CRE), methicillin-resistant *Staphylococcus aureus* (MRSA), extended-spectrum beta-lactamase (ESBL) producing *Enterobacteriaceae*, vancomycin-resistant Enterococci (VRE), multidrug-resistant (MDR) *Pseudomonas aeruginosa*, and MDR *Acinetobacter* sp. According to the latest reports from the Center for Disease Control and prevention (CDC) one out of four healthcare-associated infections will be caused by one of the above-mentioned bacteria. In 2019, the European CDC reported that multiresistance is mainly worrisome in *Acinetobacter baumannii* (43.6%), *Klebsiella pneumoniae* (15.6%), *Escherichia coli* (7.3%) and *P. aeruginosa* (3.9%) [2]. Moreover, the WHO has published the list of pathogens with critical priority in research, development and innovation of new antimicrobial therapies. This list includes carbapenem-resistant *A. baumannii*, carbapenem-resistant *P. aeruginosa* and carbapenem-resistant and ESBL-producing *Enterobacteriaceae* [3]. The worldwide dissemination of these pathogens, together with the appearance of new forms of resistance (including structures like biofilms [4]) has led to the development of the “post-antibiotic era” concept [5]. The shortage of new antibiotics in the market leads to find alternative therapies, such as phage therapy, trying to eliminate these MDR pathogens and avoid a global medical crisis.

Bacteriophages are viruses that can infect and multiply inside bacteria. They are biological systems ubiquitous in nature and very diverse from the genetic point of view; they were discovered by Felix d’Herelle in 1917. However, the first suspicions of the existence of microbes antagonistic to some bacteria were made by the British bacteriologist Frederick Twort [6]. Phages are classified as virulent or temperate depending on the biological cycle they perform, lytic or lysogenic, respectively [7]. Lysins are enzymes encoded by phages responsible for the bacterial cell wall lysis at the end of the lytic cycle and are interesting for their ability to disrupt biofilms [8]. Virulent phages are the most desirable for therapeutic use against bacterial infections. The first investigations were carried out analyzing the possible role of these viruses in medicine [9]. However, the discovery of penicillin by Alexander Fleming in 1928 and the subsequent development of clinically successful antibiotics in the following decades of the 20th century brought these lines of research to a standstill.

Phage therapy has been poorly investigated. A PubMed query in April 2021 with the terminus “phage therapy” revealed 1664 publications, most of them (96.69%) from the last 10 years. During decades, this field has suffered from a significant lack of data about many aspects such as limited knowledge about tolerance, immune response, pharmacokinetics, pharmacodynamics, and proper experiments using animal infection models. Recently, phage therapy research has been reintroduced in Western countries such as France, Belgium, the Netherlands, the United Kingdom, Switzerland, and the USA, among others. Moreover, in 2014, the US National Institute of Allergy and Infectious Diseases included phage therapy as one of seven strategies to tackle antibiotic resistance [10].

Some Eastern countries have traditionally used phage therapy under compassionate use in patients (after failure of conventional treatments) with two main Centers providing bacteriophages: The Eliava Institute in Tbilisi (Georgia), and the Ludwik Hirszfeld Institute of Immunology and Experimental Therapy in Wroclaw (Poland). Currently, the process and conditions of compassionate use are regulated by different agencies depending on the country: The Food and Drug Administration (FDA) in the USA, the Therapeutic Goods Administration (TGA) in Australia, or the European Medicines Agency (EMA) in the European Union with additional regional regulations [11].

This review summarizes the most recent studies of phage therapy in animal models and case reports of infections caused by MDR *A. baumannii*, *K. pneumoniae*, *E. coli*, and *P. aeruginosa* strains, as well as the latest clinical trials.

## 2. *Acinetobacter baumannii* and Phage Therapy

*Acinetobacter baumannii* is a Gram-negative coccobacillus extensively distributed in soil and water of natural environments. Being a nosocomial pathogen, this organism can cause pneumonia, bacteremia, meningitis or skin and soft tissue infection, among others [12]. *A. baumannii* has been isolated from community-acquired pneumonia samples [13,14] and injuries of military personnel [15,16]. Nevertheless, this pathogen is mainly associated with mechanical ventilation and burn patients [17]. Mortality rates of nosocomial infections associated with *A. baumannii* infections have ranged from 35% to 70% [18].

Importantly, this bacterium is able to rapidly become resistant to many antimicrobials. As a result, the number of multidrug and pandrug resistant *A. baumannii* strains has increased significatively in few years. Forty years ago, *A. baumannii* could be treated with traditional antibiotics. However, the current trend of resistance has introduced tigecycline and colistin to the clinical practice against this pathogen. Unfortunately, *A. baumannii* isolates resistant to tigecycline and colistin have been described.

Phage therapy against *A. baumannii* is relatively recent. The first characterization of two phages infecting this pathogen, AB1 and AB2, was done in 2010 [19,20]. Since then, many lytic phages have been characterized in vitro and in vivo. Nevertheless, the first use of phages against *A. baumannii* in humans in Western countries was reported in 2017 [21]. This successful case inspired the foundation of the Center for Innovative Phage Applications and Therapeutics (IPATH) at the University of California in 2018, whose researchers have recently published their 10 first cases of patients treated with phage therapy against different pathogens [22]. Importantly, an efficient phage therapy against *A. baumannii* could contribute to reducing morbidity and mortality in certain patient populations and make easier the complicated clinical management of infections caused by this bacterium.

### 2.1. In Vivo Phage Therapy against A. baumannii

Diverse animal models have been used to analyze the efficacy of phages, a recent review has described all different models in detail [23]. One important in vivo model is the *Galleria mellonella* larvae infection model providing a system able to bridge the gap between in vitro models and more advanced in vivo studies (mammalian), supplying initial proof-of-principle data. Mammalian models are crucial for testing safety and efficacy of phages previous to human trials, but they include the disadvantage of infrastructure, substantial costs and ethical approval.

Table 1 summarizes the in vivo studies published to date studying phage therapy against *A. baumannii* infections. Reports of phage therapy in rodents are the most frequent, they are mainly models of pneumoniae and bacteremia. All of them share similar procedures and global positive results, however, they pay attention to different issues, such as the effect of MOI or timing, the administration route, the immune response, the ability to protect against a specific strain, efficacy of cocktails versus single phages, etc. Results show that the best outcomes are obtained with phage cocktails at high MOIs administered after a short period of time post-infection. Interestingly, one of the studies reported a 20% of IgE increase after phage administration compared to the control group, however, no adverse inflammatory effect has been reported by other authors. Unfortunately, no model of *A. baumannii* infection has been performed about phage/antibiotic synergy in mammals, only two *G. mellonella* larvae models described an increase in larvae survival after a combination of phages and antibiotics. This is an important issue to be analyzed since phage use usually will be supplied with antibiotics in patients.

### 2.2. Case Reports of Phage Administration against A. baumannii

There are only five clinical cases reported to date of phage use dealing with infections caused by *A. baumannii*. This subsection describes those clinical cases in detail and Appendix A summarizes the main characteristics of phage therapy of each case.

Tan et al. described a recent case report of an 88-year-old patient suffering a chronic obstructive pulmonary disease and type-2 diabetes. Recurrent episodes of lung infections and mechanical ventilation were assessed during 2018 and 2019. The patient got infected with a carbapenem-resistant *A. baumannii* strain in May 2020 causing pneumonia. That strain was only sensitive to tigecycline and polymyxin. The reduced renal function of the patient and the low concentration of tigecycline in lungs opened the possibility to treat this patient with personalized phages. Prior to phage therapy, 50 mg of tigecycline was administrated intravenously (IV) twice a day for 6 days (d) and during the first 5 d of phage therapy. Phages were nebulized using a vibrating mesh nebulizer and administrated every 12 h (h) for 16 d, except the first 48 h with once dose per day. A gradual increment of PFU (plaque-forming unit)/dose was administered, starting with 5 × 10^6^ PFU on day 0 and reaching 5 × 10^10^ PFU on day 13. From day 6 to 10, polymyxin E was inhaled twice a day and no antibiotic was administered from day 11 to 16. Bacterial load and PFU were measured from bronchoalveolar lavage (BAL) fluid during the process. Phages were detected 1 h post-treatment (5 × 10^3^ PFU/mL) and titers were increasing up to 10^7^ PFU/mL after 15 d. Phage-resistant bacteria were isolated on days 2 and 3, but the authors suspected that those strains exhibited a lower fitness. Notably, from day 7 to the end of the treatment, no *A. baumannii* was isolated, except at day 15 with a positive culture, and the function of both lungs was improved. The patient developed a sepsis episode in August 2020 due to *Enterococcus faecium* and *Staphylococcus haemolyticus*, and also was colonized with a *P. aeruginosa*. Nevertheless, no reappearance of the carbapenem-resistant *A. baumannii* was noticed until the last test in January 2021 [44].

Wu et al. reported the case of four patients in the COVID-19-specific intensive care unit (ICU) of China receiving phage therapy as compassionate use to resolve lung carbapenem-resistant *A. baumannii* infections. Patients 2 and 4 also harbored a topical infection at the jugular incision for intubation. Phages were administered via inhalation or topically to the four critically ill men aged from 62 to 81 years old after the antibiotics failed to eradicate the *A. baumannii* infection (from 6 to 50 days before phage administration). Each treatment consisted of two doses of lytic phages spaced 1 h and patients received a total of six treatments (12 doses). Patient 1 received ΦAb124 phage via nebulization and phage-resistant strains appeared, which changed the second course of treatment including another phage, ΦAb121, forming a cocktail in combination with ΦAb124. In vitro experiments showed that those combined phages were able to suppress the recurrence of resistant bacteria, so the cocktail (and no single phages) was administered to the other patients from the beginning of their treatment to avoid resistance. The jugular incision of patient 2 was treated with the cocktail and was resolved. At the end of the treatment, patients 1 and 2 were discharged from the hospital due to the improvement of their chest radiographs. Patient 3 resolved the *A. baumannii* infection, however, a carbapenem-resistant *K. pneumoniae* infection was fatal and the patient passed away 10 d after phage therapy due to a respiratory failure. Patient 4 improved and was discharged from ICU after one week of phage treatment, nevertheless, the patient died of respiratory failure after one month [45].

A case of poly-microbial infection was described by Nir-Paz et al. A XDR *A. baumannii* and a MDR *K. pneumoniae* were isolated at day 9 after admission from the left tibia of a man aged 42 years old who was in the trauma unit due to a bicondylar tibial plateau fracture and a right distal femoral fracture. The patient was treated with 6 weeks of piperacillin/tazobactam, and 8 weeks of meropenem and colistin, however, the XDR AbKT722 strain was isolated and exhibited resistance to colistin and carbapenem. Interestingly, the inflammatory markers were almost normal at that point. Amputation was the following standard option, so phage therapy was considered. Five doses containing 1 mL of a mix of ΦAbKT21phi3 and ΦKpKT21phi1 phages (targeting *A. baumannii* and *K. pneumoniae*, respectively) were administered IV (5 × 10^7^ PFU/mL) in combination with meropenem and colistin along 5 d and no adverse effects were found. One week later, a second treatment of 6 d was administered and signs of recovery were achieved. Finally, a follow-up of eight months failed to isolate *Acinetobacter* and *Klebsiella* from the patient [46].

LaVergne et al. described the case of a 77-year-old man who suffered an infection after a craniectomy caused by a MDR *A. baumannii* strain resistant to all antibiotics, although some isolates were sensitive to colistin. Phage therapy was initiated with a screening of a collection containing 104 phages infecting *A. baumannii*. Only five phages lysed the strain and the one with the highest lytic effect was chosen for therapy. The first dose was administered IV at day 12 after admission with 4 mL of 2.14 × 10^7^ PFU/mL and a total of 96 doses were administered during 8 d. After 5 min post-treatment, 110 PFU/mL was found in serum, nevertheless, a rapid clearance of phages occurred in all treatments (after 10 min from phage delivery). Almost 2 h after the first phage administration, the patient experienced a brief hypotension, although no vasopressors were required. Fever and leukocytosis persisted and the patient’s family decided to remove intubation and other cares at day 19 and the patient passed away on day 20 [47].

Schooley et al. described the case of a 68-year-old man suffering diabetes and necrotizing pancreatitis which was complicated by an MDR *A. baumannii*-infected pseudocyst. The antibiotic therapy did not resolve the infection and phages were considered as an option. Three different cocktails were administered: ΦPC (AC4, C1P12, C2P21, C2P24) from day 109 during 18 weeks through percutaneous catheters, ΦIV (AB-Navy1, AB-Navy4, AB-Navy71, and AB-Navy97) from day 111 during 16 weeks through IV administration, and ΦIVB (AB-Navy71, AbTP3Φ1) from day 221 during 2 weeks via IV to eliminate a phage-resistant *A. baumannii* isolated 8 d after the first phage treatment. Phage therapy was combined with minocycline, fluconazole and meropenem. Remarkably, that treatment resulted in MDR *A. baumannii* eradication and the clinical recovery of the patient. The authors also studied the phage concentration after IV administration and found that, after an IV challenge of 5 × 10^9^ PFU, the levels in serum decreased dramatically: 1.8 × 10^4^ PFU/mL by 5 min, 4.4 × 10^3^ PFU/mL by 30 min, 3.3 × 10^2^ PFU/mL by 60 min, and 20 PFU/mL by 120 min post-injection; suggesting that phage neutralization may decrease activity in plasma [21].

Case reports described against *A. baumannii* share a similar trend to utilize phage cocktails combined with antibiotics in more than one dose via IV or nebulized, depending on the infection. Phage therapy has obtained a global positive impact according to the reduction of the bacterial burden, however, it has not correlated with recovery in all cases (Appendix A).

## 3. Phage Therapy against Infections Caused by *K. pneumoniae*

*K. pneumoniae* is a Gram-negative bacillus that belongs to the *Enterobacteriaceae* family and is an opportunistic pathogen responsible for a wide range of nosocomial infections, including urinary tract infections (UTIs), respiratory tract infections, blood-stream-associated infections, surgical-site infections, and also is responsible for liver abscesses in community acquired infections [11]. Importantly, the intestine of patients acts as a reservoir for *Klebsiella pneumoniae* within a hospital [2]. This pathogen causes severe morbidity and mortality in ICUs, premature ICUs and medical, pediatric and surgical wards. *K. pneumoniae* strains are intrinsically resistant to penicillins and can acquire resistance to third- and fourth-generation cephalosporins [12]. Third-generation cephalosporins have been used as monotherapy and in combination with aminoglycosides against *K. pneumoniae* infections. Fosfomycin is mainly used in UTIs. This practice accelerated the emergence of fluoroquinolone-resistant and ESBL-producing *K. pneumoniae* strains which are treated with carbapenems. Carbapenem use has also increased the appearance of CRE bacteria which are resistant to carbapenems, penicillins, cephalosporins, fluoroquinolones, and aminoglycosides. Carbapenemases are a class of enzymes that can be transferred rapidly through conjugative plasmids among the Enterobacteriaceae family. Treatment options against carbapenem-resistant *K. pneumoniae* are limited to colistin and tigecycline. Unfortunately, resistance to these antibiotics has also been described. Moreover, carbapenem-resistant *K. pneumoniae* is associated with 50% of mortality rates in Europe [13]. A systematic review and meta-analysis of mortality concluded that patients infected with carbapenem-resistant *K. pneumoniae* strains exhibited higher mortality rates (42.14%) than those infected with carbapenem-sensitive strains (21.16%), especially in association with bloodstream infection, ICU admission, and solid organ transplantation [13].

Studies of phage therapy against *K. pneumoniae* are not scarce. In 2020, Herridge et al. reviewed therapeutic phages against MDR *Klebsiella* spp. summarizing in vitro and in vivo experiments until 2019 [48]. In this scenario, this section will describe the additional recent works published to date of phage therapy against *K. pneumoniae* in animal models and case reports.

### 3.1. In Vivo Phage Therapy Experiments against K. pneumoniae

Recent in vivo studies of phage therapy against *K. pneumoniae* are summarized in Table 2. Pneumonia and sepsis were the most studied infections and bacteriophages usually were administered by the same route at different times. Curiously, no UTI model of infection has been recently reported, despite the fact that *K. pneumoniae* is typically isolated from UTIs in humans. Diverse murine studies showed that survival rates of mice depended more on time of phage administration than on dose. More studies using cocktails of phages and combinations of phages with antibiotics are needed in order to determine the optimal strategy to perform successful clinical trials.

### 3.2. Case Reports Using Phage Therapy against K. pneumoniae

Recent case reports of phage use against *K. pneumoniae* reveal the potential usefulness of this approach in clinical practice, mainly in chronic infections.

Qin et al. reported the case of a 66-year-old man with a multifocal UTI caused by an MDR *K. pneumoniae* strain which was treated with a cocktail of phages. This patient suffered from UTIs caused by *K. pneumoniae* for 12 years and his bladder mucosa become hyperemic with a local ulceration. Kp4173 and Kp0344 bacterial strains were isolated from the patient urine and 21 total *K. pneumoniae* strains belonging to ST15 were isolated from the patient through the treatment. ΦJD902 was used for the first phage therapy treatment during two weeks without antibiotics through a renal pelvis effusion. After phage treatment, five urine isolates were resistant to ΦJD902. Consequently, the next therapy used a cocktail of two phages, ΦJD902 and ΦJD905. However, phage-resistant isolates were also found and another phage cocktail was administered (ΦJD905, ΦJD907 and ΦJD908) via bladder irrigation. Cultures remained positive for bacteria and piperacillin/tazobactam was administered. The authors began to irrigate the bladder and renal pelvis via kidney and bladder with a new cocktail (ΦJD902, ΦJD905, ΦJD908, and ΦJD910) combined with the two antibiotics and the last 10 d without antibiotics. Finally, the patient recovered his bladder healthy and no MDR *K. pneumoniae* was isolated in a follow-up of two months [57].

Cano et al. reported the case of a phage treatment to rescue a prosthetic knee infected by *K. pneumoniae*. Phage KpJH46Φ2 was administered to a 62-year-old diabetic man with total knee arthroplasty. In 2008, a primary infection with *S. epidermidis* appeared, followed by *S. pyogenes* and *S. aureus* infection. In 2018, the patient got infected with *E. faecalis* which was eradicated with antibiotics. In 2019, the patient was positive for *K. pneumoniae* complex and treated with meropenem and minocycline, but after one month, infection reappeared and amputation was recommended. Phage therapy was considered at that point and was administered with minocycline (oral 100 mg/12 h). The patient received a total of 40 IV doses of daily infusions containing 6 × 10^10^ PFU. In vitro experiments demonstrated an antibiofilm effect of the phage. The patient experienced a rapid improvement after 48 h of the treatment and inflammatory markers decreased. Anti-phage antibodies were also analyzed and, interestingly, results showed no change over time [58].

Corbellino et al. presented a case of oral and intra-rectal phage administration against an MDR *K. pneumoniae* strain containing a KPC carbapenemase. In 2017, a woman aged 57 years old suffering Chron’s disease was colonized by an MDR *K. pneumoniae* in the gastrointestinal tract, the urine tract and an internal device. After different clinical episodes of infection, the patient ended in sepsis with positive blood cultures for MDR *K. pneumoniae* and was treated with ceftazidime-avibactam which resolved the situation. One week post-recovery, positive cultures were encountered in urine and in a rectal swab. Phage therapy was considered and, before starting, the patient developed fever that was reduced using ceftazidime-avibactam, although her urine was positive for the same strain. Phage therapy was administered, and no adverse effects were recorded. Eradication of *K. pneumoniae* was achieved and molecular screening confirmed that no *Klebsiella* was present in rectal swabs in December 2018 [59].

Bao et al. successfully treated a recurrent UTI caused by an XDR *K. pneumoniae* strain using phage therapy. A 63-year-old patient with a chronic UTI was treated with a phage cocktail composed of phages SZ-1, SZ-2, SZ-3, SZ-6 and SZ-8 (5 × 10^8^ PFU/mL) against the *K. pneumoniae* isolate CX7224 ST11. However, *K. pneumoniae* became resistant to the first cocktail and new resistant isolates emerged also after a second and a third round of phages. A last cocktail was applied (Kp152, K154, Kp155, Kp164, Kp6377, and HD001) and that time was combined with the non-active antibiotic trimethoprim-sulfamethoxazole. The synergistic effect was studied in vitro before the application to the patient and results showed that *K. pneumoniae* became resistant to the phage cocktail but was eradicated by the combination with the antibiotic. The patient received the drug (800–160 mg) combined with the phage irrigation of the bladder and using this strategy *K. pneumoniae* was eradicated with no signs of recurrence in a follow-up of 6 months [60].

Rostkowska et al. described the case of a 60-year-old patient with a kidney transplantation and a chronic UTI. An MDR *K. pneumoniae* strain was firstly treated with meropenem and colistin. After 10 severe UTIs, phage therapy was suggested. At the fifth day of phage administration, the temperature of the patient reached 37.2 °C. A number of cysts were found in the patient´s kidney and meropenem was administered. Recurrent episodes were treated with meropenem and phage therapy. However, the patient’s left kidney was planned to be removed due to the presence of infected cysts and the recurrence of infections. Finally, the patient was discharged home 4 d after the surgery and no signs of infection were found after 5.5 years from the surgery [61].

Similarly, Kuipers et al. described the phage treatment of a 58-year-old patient suffering a recurrent UTI and an epididymitis due to an ESBL *K. pneumoniae* after renal transplant. One week post-transplant, the patient was infected by a *K. pneumoniae* strain susceptible to meropenem and amikacin. The patient started with UTI recurrence that could not be resolved after seven times of meropenem therapy (from 10 d to 4 weeks of treatment). Phage therapy was considered and phages from the Eliava Institute in Tbilisi (Georgia) were administered according to the instructions. Briefly, they recommended two vials of phages per day orally and another one via intravesical every second day during two months. The phage treatment was coincident with meropenem administration to treat his epididymitis and no adverse effects were observed. As a result, the urethral symptoms decreased from the beginning of phage administration and no bacterial isolate was found after 14 months of follow-up [62].

Rubalskii et al. described, among others, the case of a 40-year-old male patient who developed a lung infection during immunosuppression after heart transplantation. The phage treatment was against a pandrug-resistant *K. pneumoniae* strain and consisted of 2 d of one dose of 2 mL of inhaled phages and 18 mL of nasogastric administration containing 1 × 10^8^ PFU/mL of phages KPV811 and KPV15 followed by 2 d of the same treatment administered twice a day. Antibiotic therapy was also used. Importantly, *K. pneumoniae* was not detected in the BAL, although isolates were encountered in stool samples after treatment. Fortunately, the isolated strain was susceptible to antibiotics, indicating that no additional phage therapy was necessary [63].

These seven cases describe successful treatments of phage therapy mainly against UTI infections caused by MDR *K. pneumoniae* strains. Antibiotics combined with cocktails of phages were administered by diverse routes, such as IV, intrarectal, ladder irrigation, oral and intravesical against the UTI infections (Appendix A).

## 4. Phage Therapy against *E. coli*

*E. coli* is a Gram-negative bacillus belonging to the *Enterobacteriaceae* family. Its natural habitat is the intestine of humans and animals and is an excellent indicator of fecal contamination in water. *E. coli* can cause infections in humans in different parts of their anatomy and physiology, being the leading cause of UTI (ECSEP serotype) in about 70–80% of cases due to its ability to develop fimbriae and adhesins and the production of the siderophore aerobactin. It is also responsible for bacteremia (ECSEP serotype) and meningitis (K1 serotype), and is also an etiological agent of gastrointestinal tract infections. The main serotypes causing gastrointestinal infections are Enteropathogenic *E. coli* (ECEP), responsible for watery diarrhea in children; Enterotoxigenic *E. coli*, associated with traveler’s diarrhea and food poisoning due to the production of thermolabile and thermostable TSa and TSb toxins, respectively; Enteroinvasive *E. coli* (IEEC), causing bacillary dysentery; Enterohemorrhagic *E. coli* (EHEC) being 0157:H7 the most important serotype, responsible for hemorrhagic colitis and uremic and hemolytic syndrome due to the production of Shiga toxin or verotoxins; finally, Enteroaggregative *E. coli* (ECET), affecting children in the form of watery diarrhea. *E. coli* is a very frequent pathogen at both community and hospital level being one of the pathogens responsible for nosocomial infections. This organism can be resistant to different antibiotics such as quinolones or cotrimoxazole, but it stands out mainly for its ability to carry ESBL enzymes [64]. Phage therapy represents an interesting alternative for the prevention and treatment of infections caused by this bacterium. In this section we focus on phage therapy studies against *E. coli* published in the last 10 years, with only two case reports published during this period.

### 4.1. In Vivo Models of Phage Therapy against E. coli

Due to the presence of intestinal *E. coli*, phage therapy has been evaluated in vivo not only to be used as therapy, but also as prophylaxis. Table 3 reviews the in vivo studies of phage therapy against *E. coli* and we can distinguish between two types of studies.

One type is focused on studying the prophylactic or therapeutic phage use against pathogenic strains at the intestinal level. Importantly, cocktails of phages have shown to be more effective for prophylaxis than single phages. Good results have been reported using different phage administrations (drinking water, oral injection or vegetable capsules), although in one case, the burden of the target bacteria increased again after withdrawing phages.

The other studies are focused on murine models of infection to resolve sepsis or pneumonia similarly to the experiments reviewed for *A. baumannii*, *K. pneumoniae* or *P. aeruginosa*. Curiously, one of the studies analyzed the kinetics of phages and results demonstrated that most of bacteriophages were accumulated in the murine spleen.

### 4.2. Case Reports Using Phage Therapy against E. coli

Rubalski et al. described the case of a 66-year-old-patient with a wound swab infected by *E. coli* [63]. Phage therapy was administered when conventional antibiotic therapy could not improve the inflammation and *E. coli* was recurrently isolated. In total, 4 mL of a cocktail of two phages, ECD7 and V18, were administered (4 × 10^10^ PFU/mL) intraoperatively mixed with fibrin glue. Clindamycin was combined with phages (600 mg three times per day). Finally, phage therapy succeeded, and *E. coli* was no longer detected after phage administration.

A 56-year-old patient was treated with phage therapy to deal with a recurrent UTI due to an ESBL *E. coli* after a liver transplantation and immunosuppression. The patient also suffered from prostatitis, chronic kidney disease and kidney stones. A cocktail of four phages named UCS1 was administered (10^9^ PFU/mL) IV every 12 h for 2 weeks in combination with ertapenem and, during 12 weeks of follow-up, symptomatic UTIs disappeared although urine cultures were positive [22].

These two case reports show that phage use could be desirable in recurrent or chronic infections caused by *E. coli*.

## 5. *Pseudomonas aeruginosa* and Phage Therapy

*P. aeruginosa* is a prevalent Gram-negative bacillus found in water, oil, different surfaces, medical devices which can be isolated from plants, animals, and humans. Currently, *P. aeruginosa* constitutes one of the main opportunistic pathogens which can cause a wide variety of nosocomial, acute, and chronic infections (including pneumonia, septicemia, urinary tract and surgery site infections), especially in immunocompromised individuals. Cystic fibrosis (CF) patients are particularly vulnerable to *P. aeruginosa* infections being one of the major causes of mortality and morbidity [74,75]. This pathogen causes chronic lung infection, deterioration of pulmonary function and, in the worst cases, death, and colonizes 30% of children and even 80% of older 25-year-old adults suffering CF [76].

This bacterium is capable of invading host cells and avoiding host defenses due to an arsenal of virulent secreted factors, such as proteases, elastases, pyocyanins, exotoxin A, phospholipases, exoenzymes, and cell-associated factors (lipopolysaccharides, flagella and pili) [77]. Moreover, *P. aeruginosa* contains a low permeable outer membrane and multiple transport systems, providing an innate resistance to many antibiotics [78]. In addition to its innate resistance, *P. aeruginosa* is able to develop resistance to almost all available antimicrobials [79] showing resistance against several antimicrobials including fluoroquinolones, β-lactams, and aminoglycosides, with MDR and XDR varieties [80]. The main mechanisms conferring resistance in MDR *P. aeruginosa* consist of alterations in porin channels, efflux pumps, target modifications and β-lactamases (for instance, AmpC and carbapenemases) [81]. Antimicrobial resistance can be acquired by selection of mutations in chromosomal genes or by horizontal uptake of resistance determinants. Furthermore, failure of antimicrobial treatments has also been associated with the formation of biofilms [82].

Owing to the absence of appropriate and effective treatments, researchers are looking for new ways of inhibiting MDR and XDR *P. aeruginosa* strains, being phage therapy one of the most promising methods. Phages against *P. aeruginosa* have been isolated from hospital sewage, seawater, ponds, rivers and wastewater-treatment plants [80]. The first bacteriophages against the *Pseudomonas* genus were described in the middle of the 20th century [83,84]. In 2015, Pires et al. reviewed 137 completely sequenced *Pseudomonas* phage genomes that were registered in public databases [77]. This section reviews in vivo studies and case reports published since 2015 to date.

### 5.1. Phage Therapy against MDR/XDR P. aeruginosa: In Vivo Studies

Several studies have been conducted on the use of bacteriophages using in vivo models in order to evaluate the safety and efficacy of phages to neutralize clinical pathogens. Table 4 summarizes recent results. It shows that *G. mellonella* models of infection demonstrated no toxicity of phages and excellent results in prophylaxis. Recent murine studies mainly investigated the effect of phage therapy in pneumonia using models with intranasal (IN) or aerosolized inoculations. MOIs higher than 10 showed high survival rates and only one murine study combined phages with antibiotics resulting in better results. Curiously, phage administration was reported to be effective even when it was administered after long periods post-infection, which is dissimilar to the results published for *A. baumannii* and *K. pneumoniae*. On the other hand, one innovative CF zebrafish model has been developed to examine phage therapy against *P. aeruginosa* infections in CF patients [74]. Zebrafish CFTR channel conforms a similar structure to the human CFTR and embryos with CFTR knockdown present sensitivity to infections caused by *P. aeruginosa*, as it occurs in CF patients. Although zebrafish do not have lungs, they have mucin, the proteins overexpressed in the lungs of CF patients, which are highly similar to human mucins. All these peculiarities along with the evidence of development of *P. aeruginosa* microcolonies (biofilm precursors) in zebrafish makes it a good model of CF.

### 5.2. Case Reports of Phage Administration against P. aeruginosa

In recent years, most cases of bacteriophage therapy have been administered in combination with antibiotics, however, there are some examples of phage therapy in humans administering single treatment. All these cases are summarized in Table 3. 

Jennes et al. described the case of a 61-year-old man who developed septicemia caused by a colistin-only-sensitive *P. aeruginosa* strain [91]. Unfortunately, the patient also suffered from acute kidney injury, so drug administration was discontinued to prevent further kidney damage. Due to the colistin nephrotoxicity, phage therapy was initiated IV with the phage cocktail BFC1 every 6 h and his wounds were irrigated with phages every 8 h during 10 d. Immediately, blood cultures turned negative and after a few days, the function of his kidneys was fully recovered.

Duplessis et al. described another medical case, a 2-year-old child who suffered from DiGeorge syndrome and a complex congenital heart disease [92]. The patient experienced recalcitrant *P. aeruginosa* bacteremia being treated with multiple antibiotics (meropenem, tobramycin, aztreonam, colistin and polymyxin B) to which the organism initially was susceptible. Eventually, the patient exhibited adverse reactions or resistance to all previously mentioned antibiotics. Given the lack of antibiotic options, the clinical team initiated IV phage therapy to target *P. aeruginosa*. Bacteriophages were administered to the patient at a dose of 3.5 × 10^5^ PFU every 6 h limited by the number of endotoxin units administered according to FDA guidelines (5/kg per h). The patient tolerated the first six doses of phage therapy, but phage administration was suspended after decompensation for anaphylaxis attributed to progressive heart failure. The patient resumed phage therapy and results from blood cultures, that had reverted to positive, reverted again to sterile for several days coinciding with clinical improvement.

Tkhilaishvili et al. described the clinical case of an 80-year-old woman with diabetes mellitus type 2, chronic kidney failure and diagnosis of relapsing knee periprosthetic joint infection with a MDR *P. aeruginosa* strains [93]. The patient was infected with two different strains: one was resistant to all antibiotics except colistin, and the other was only susceptible to colistin and ceftazidime. The knee prothesis was explanted and in the surgical site was deposited an antibiotic-loaded cement spacer containing 1 g of gentamicin and 1 g of clindamycin per 40 g of poly (methyl methacrylate). During surgery, the patient started with phage therapy against the two strains. Purified bacteriophages were applied locally during surgery and after surgery (10^8^ PFU/mL) every 8 h through drains for 5 d. Importantly, on days 3, 4 and 5 from phage therapy, drainage fluids were collected, and no bacterium was isolated.

Law et al. described another clinical case in which antibiotic and bacteriophages were combined [94]. A 26-year-old female patient with CF in the lung transplant waitlist was admitted in hospital with pulmonary exacerbation leading to acute-on-chronic respiratory failure due to colonization with two strains of MDR *P. aeruginosa*: one strain sensitive to colistin and the other sensitive to meropenem and piperacillin/tazobactam. Firstly, she was treated with antibiotics during 4 weeks: piperacillin-tazobactam for the first two weeks, carbapenem for the last two weeks, and colistin and azithromycin for the entire period. At the end of the antibiotic treatment, she exhibited a worse medical status and was transitioned to inhaled colistin as suppressive therapy and worsened over the following week with progressive respiratory and renal failure. As a result, she was out of the transplant waitlist. At this point, physicians started phage therapy via IV and AB-PA01 phage was administered every 6 h (4 × 10^9^ PFU/mL) for 8 weeks. At the beginning of phage therapy, colistin was discontinued and the patient received concomitant ciprofloxacin and piperacillin-tazobactam for 3 weeks. During the last part of phage therapy, ciprofloxacin was discontinued and changed to doripenem based on updated *P. aeruginosa* sensitivity profiles. After 100 d since phage therapy was finished, she did not suffer recurrence infections with *Pseudomonas* or CF exacerbation, furthermore, the patient underwent a successful bilateral lung transplantation nine months later.

Aslam et al. reported two similar cases of two patients that were infected with MDR *P. aeruginosa* and due to the inability of eradicating the infection with conventional antibiotics in both cases, combined treatment was employed [95]. The first patient was a 67-year-old man who underwent bilateral lung transplant for hypersensitivity pneumonitis. His post-transplant course was complicated due to multiple medical issues, including *P. aeruginosa* pneumonia suffering two different episodes of bacterial infection and receiving combined treatment with antibiotics and bacteriophages. During the first episode, he received IV and nebulized AB-PA01 (4 × 10^9^ PFU/mL) during two weeks as an adjuvant to antibiotic treatment with piperacillin-tazobactam and colistin. Antibiotic treatment was stopped on day 18 and phage therapy was prolonged one additional week in an attempt of repopulating the airways with normal respiratory flora. At day 21, cultures did not include *P. aeruginosa*. Nevertheless, at day 46, clinical conditions of the patient worsened and respiratory cultures were positive for MDR *P. aeruginosa* restarting combined treatment with antibiotics (piperacillin-tazobactam, tobramycin and inhaled colistin) and bacteriophages. Phage therapy consisted of AB-PA01-m1 (5 × 10^9^ PFU/mL) and a phage cocktail (1 × 10^9^ PFU/mL). Once the combined treatment was administered, a complete clinical resolution of pneumonia was achieved. The patient received bacteriophages for almost 8 additional weeks as prophylactic therapy against a possible recurrence. No infection reappeared in the next three months demonstrating the potential efficacy of preventive approach in humans. The second patient was a 57-year-old woman suffering recurrent *Pseudomonas* infections with a bronchiectasis which was colonized with an MDR *P. aeruginosa* strain sensitive only to colistin. Due to the incapacity to eradicate *P. aeruginosa* from the respiratory tract, phage therapy combined with antibiotics was tested. The patient was treated with a 4-week course of IV AB-PA01 (4 × 10^9^ PFU/mL) and inhaled colistin. Finally, the results showed that no additional *P. aeruginosa* was cultured from respiratory samples.

Maddocks et al. reported the first case in which phage therapy, combined with other antimicrobials, was used clinically as treatment against an extensive, necrotizing, pulmonary pseudomonal infection with resolution of the *P. aeruginosa* infection [96]. A 77-year-old woman underwent a right posterolateral mini thoracotomy for resection of right lower lobe adenocarcinoma with mediastinal node sampling developing pneumonia and empyema caused by *P. aeruginosa* sensitive to piperacillin-tazobactam, ciprofloxacin and meropenem. The patient was treated with IV meropenem 1 g three times per day. Clinical conditions became worse and direct pleural swabs were positive to *P. aeruginosa* and resistant to meropenem, imipenem and piperacillin-tazobactam. Meropenem was ceased and a treatment based on ciprofloxacin and gentamicin was IV administrated. Phage therapy with AB-PA01 (1 × 10^9^ PFU/mL) combined with antibiotics was started IV and nebulized twice a day. Fortunately, after 4 d of combined treatment, the patient was culture-negative and remained negative after six months of follow-up.

Rubalskii et al. described the case of a 13-year-old male patient that developed a *P. aeruginosa* infected-thoracotomy wound two months after a double lung transplantation due to CF [63]. *P. aeruginosa* strains were not eradicated with surgical debridement, vacuum-assisted therapy, and continuous antibiotic therapy. After continuous isolation of *P. aeruginosa* and deterioration of his clinical conditions, phage therapy was administered consisting of PA5 and PA10 administration (4 × 10^10^ PFU/mL) combined with IV administration of colistin (twice per day), and ceftazidime and avibactam (three times per day). As a result of this combined treatment, no *P. aeruginosa* was isolated indicating clinical success.

Chan et al. described the case of a 76-year-old male patient that underwent aortic arch replacement surgery with Dacron graft for an aortic aneurysm. One year after surgery the patient developed a recurrent *P. aeruginosa* infection which was treated with ciprofloxacin and ceftazidime. However, the bacterial strain become resistant to those antibiotics [97]. Phage therapy was proved with OMKO1 bacteriophage (10^7^ PFU/mL) combined with ceftazidime (0.2 g/mL) and was applied into the mediastinal fistula. A period after, ceftazidime was discontinued, and the patient did not manifest any evidence of recurrent infection.

These case reports against *P. aeruginosa* describe the resolution of different infections using a mix of phages administered with antibiotics mainly via IV and nebulized. Chronic infections and patients with comorbidities are the most frequent cases.

## 6. Clinical Trials

The number of clinical trials testing phage therapy against the four reported bacteria is scarce compared to the number of in vivo experiments and case reports. Most of them are phase-I/II clinical trials studying safety instead of efficacy. Some have been performed specifically against single bacteria while other focus on a concrete infection disease caused by different bacteria.

### 6.1. Clinical Trials against Poly-Infections

Here, we first mention seven trials using phage therapy against more than one bacterial species, some lack published results. Aleshkin et al. reported a small clinical trial using a cocktail of eight bacteriophages to study the effect on therapy and prophylaxis against MDR *A. baumannii*, *K. pneumoniae*, *P. aeruginosa* and *S. aureus* in patients in an ICU. No toxic effects were observed after the administration of phages. Phages were administered intragastrically in a period of 3 d to 14 patients (20 mL containing 10^8^ PFU/mL) and the bacterial load decreased at 24 h after the treatment [98]. Bochkareva et al. published different clinical cases of 42 patients in a neurological ICU of Russia with a prolonged lung ventilation as prophylactic use. A cocktail of phages against *A. baumannii*, *K. pneumoniae*, *P. aeruginosa* and *S. aureus* was used containing two phages per type of bacteria. Patients treated with phages showed a range of 54–62.5% of infected loci resolved. Importantly, anti-phage IgGs were developed after the oral administration of phages and subsequent phage administrations did not eradicate bacteria. The authors concluded that repeated doses with the same phage could be not effective due to the immune response [99]. The aim of the clinical trial PhagoBurn (NCT02116010) (*n* = 27 patients) was to evaluate phage therapy as treatment against *E. coli* and *P. aeruginosa* wound infections in burned patients. While they found that bacterial burden of *P. aeruginosa* in most infected wounds was successfully reduced at the end of phage therapy [100], the median time to achieve this endpoint was significantly longer for those in the phage therapy group (143 h) than for those in the standard care group (47 h) and the clinical trial was cancelled before getting finished. The authors found that after manufacturing, the titer of the phage cocktail decreased and the patients received lower concentrations than expected (from 10^6^ to 10^2^ PFU/mL).

Another clinical trial which is currently recruiting patients (NCT04803708) (*n* = 26 participants) aims to evaluate the safety of phage therapy (TP-102) as treatment of diabetic foot ulcers against different pathogens, including *P. aeruginosa*, *A. baumannii* and *S. aureus*. Bacteriophage will be administered topically containing 10^9^ PFU/mL of phages. Another trial (NCT04815798) (*n* = 69 participants) intends to evaluate safety, efficacy, and tolerability of phage therapy (BACTELIDE) combined with the standard care in pressure ulcer colonized by *P. aeruginosa*, *K. pneumoniae* or *S. aureus*. BACTELIDE is a cocktail that contains 14 bacteriophages encapsulated in a biodegradable polymer. The clinical trial NCT04323475 (*n* = 12 participants) is focused on safety and efficacy of the phage cocktail-SPK combined with the standard of care (xeroform and kenacomb topical antibiotic cream) as treatment and prevention of burns susceptible to be infected by *P. aeruginosa*, *S. aureus* and *K. pneumoniae* species. Phage therapy consists of dosage-metered airless spray containing a cocktail of 14 phages at a concentration of 1.4 × 10^8^ PFU/mL for an effective dosage of 2.5 × 10^5^ PFU/cm^2^ of burned area. Recently, as a consequence of the Coronavirus pandemic (COVID-19), a clinical trial (NCT04636554) has been developed to achieve personalized phage treatment in COVID-19 patients with bacterial co-infections causing pneumonia or bacteremia/septicemia. The aim of this clinical trial is to determine the viability of developing a personalized bacteriophage treatment for COVID-19 patients who suffer pneumonia, bacteremia, or septicemia due to *P. aeruginosa*, *Acinetobacter* or *S. aureus*, and to evaluate safety of phage administration combined with conventional antimicrobial treatments.

Leitner et al. reported a randomized, placebo-controlled clinical trial in phase II/III (NCT03140085) (*n* = 97 patients) that recruited men suffering from UTIs who were scheduled for transurethral resection of the prostate with isolates of *Enterococcus* spp., *E. coli*, *Proteus mirabilis*, *P. aeruginosa*, *Staphylococcus* spp., and *Streptococcus* spp. A group of patients received the commercially available Pyophage (*n* = 37), another group was treated with antibiotics (*n* = 38), and a third group was a placebo group (*n* = 38). Intravesical phage therapy was non-inferior to antibiotic treatment but was not superior to placebo bladder irrigation. Fortunately, no concerns about safety were reported.

### 6.2. Clinical Trials against Mono-Infections

#### 6.2.1. *K. pneumoniae*

Patel et al. described a prospective study of 48 patients with nonhealing wounds during 6 weeks, including diabetic and hypertensive patients to administer phage therapy against different bacterial infections, including *K. pneumoniae* (12.5%). A mono-phage therapy was applied to mono-bacterial infections and a cocktail of specific phages was administered in case of mixed bacterial infections. Phages were applied topically every 48 h. A follow-up of 3 months was carried out and 39 out of 48 patients (81.2%) reached cure. Curiously, infections with *K. pneumoniae* showed a slow healing compared to others [101].

#### 6.2.2. *E. coli*

Sarker et al. conducted a randomized Phase 1 clinical trial evaluating safety of a T4 phage cocktail in Bangladeshi children [102]. This phage was compared with a commercial preparation of *E. coli*-*Proteus* bacteriophage from Microgen (Russia). In both cases no adverse effects were observed, both preparations were administered orally, and fecal microbiota variability was assessed over 7 d using 16S rRNA sequencing. A significant variability of the microbiota was encountered, this variability in the microbiota was also reported in 71 pediatric patients being treated for diarrhea by rehydration versus 38 patients receiving the preparation with T4 phages. In 2016, Sarker et al. developed another trial [103]. The cocktail of T4 and T7 phages from Microgen ColiProteus was compared with another commercial preparation of T4-like cocktail from Nestlé Research Center and with the standard treatment which is the use of oral rehydration sera. The trial was randomized and double-blinded enrolling 120 hospitalized children aged 4–60 months with ETEC and EPEC diarrhea. No adverse effects related to the oral administration of the phages were observed, however, no significant improvement was shown with respect to the standard treatment. The authors pointed that low titers of phages detected in the stool of patients and the low number of cases of enrolment could be the explanation. In another study, Sarker et al. sequenced a mixture of 99 T4 phages. Nine bacteriophages were administered orally in a cocktail to 15 healthy adult volunteers from Bangladesh at doses of 3 × 10^9^ and 3 × 10^7^ PFU, and placebo. The aim of this study was to analyze the impact and safety of phage administration in humans. The presence of *E. coli* was not detected in the initial stool analysis and no adverse effects were reported showing normal liver function, renal function and hematological analysis. Importantly, no impact was detected on fecal microbiota [104].

Another trial (NCT0419148) studied safety, pharmacokinetics, pharmacodynamics and efficacy of the LBP-EC01 cocktail versus placebo consisting of Ringer’s solution for injection in 30 patients who have had at least one episode of *E. coli* UTI or have permanent or intermittent urinary catheters, including those with asymptomatic bacteriuria caused by *E. coli* colonization with a count of more than 10^3^ CFU (colony-forming units)/mL bacteria. The trial was blinded and randomized but the results have not been published yet, although it has been concluded. Another clinical trial (NCT3808103) (*n* = 30 patients) which is still in the patient recruitment phase aims to evaluate safety of the EcoActive bacteriophage preparation against adherent-invasive *E. coli* in patients with Crohn’s disease. Adherent-invasive *E. coli* may be responsible (among other causes) for part of the inflammatory symptoms of Crohn’s disease, so the use of bacteriophages against this bacterium could improve the prognosis of these patients. In contrast to the use of antibiotics, phages would not affect the rest of the intestinal microbiota. The trial is a phase I randomized trial carried out at Mount Sinai Hospital and compares the effect of the aforementioned phage cocktail against a placebo preparation.

#### 6.2.3. *P. aeruginosa*

The aim of a recent clinical trial (NCT04596319) (*n* = 48 participants) is to evaluate safety and tolerability of the inhaled administration of phage AP-PA02 in patients that suffer from chronic pulmonary *P. aeruginosa* infections and CF. This trial is composed by two different parts. The first part will analyze the effect of single doses of phage therapy at three ascending dose levels, while the second part is going to evaluate the safety and efficacy of multiple doses of phage therapy in each of the three ascending dose levels groups. Control group will receive an inactive placebo dose administered via inhalation. The trial NCT04684641 (*n* = 36 participants) is focused on the reduction of sputum bacterial burden in CF patients infected with *P. aeruginosa* with the administration of YPT-01 phage. Moreover, another purpose of this study that has finished its recruiting process is to demonstrate the clinical efficacy and safety of the inhaled or nebulized phage therapy YPT-01. In the list of clinical trials associated with phage therapy and *P. aeruginosa* infections, we can find an expanded access program (NCT03395743), no longer available. The aim of this trial was to allow physicians to provide treatment with AB-PA01, an investigational bacteriophage therapeutic for treatment of *P. aeruginosa* infections, for patients with serious or immediately life-threatening infections for which alternative treatments are not currently available. Currently, there are no posted results available for this trial. Finally, an in vitro clinical trial (NCT01818206) (*n* = 59 participants) intends to evaluate the efficacy of a cocktail of bacteriophages in infecting *P. aeruginosa* strains present in sputum samples of patients. However, no results have been published yet.

Clinical trials reviewed here have a modest *n* of participants because they are phase 0, I or II. To date, only one clinical trial in phase II/III has been reported for these bacteria using phage therapy. An important issue in clinical trials is the stability of phages through the study, since a decrease of titer compromises the goal of the entire trial, as it happened in PhagoBurn.

## 7. Discussion

The exponentially incremented number of studies related with phage therapy in the last decade highlights the need for alternative therapies to antibiotics due to multiresistance. Phage therapy is a realistic alternative to antibiotics for humans, although there are some concerns that remain unclear. Specificity of phages is a peculiar characteristic which enables the targeting of desired bacteria without disturbing other bacteria avoiding dysbacteriosis. Unfortunately, the narrow host range of bacteriophages forces to design cocktails with more than one tailored phage against a single strain which lengthens the development process. This is an important issue since the lack of available pre-isolated and characterized phages leads to isolate new phages again and again generating a substantial delay from request to administration. As a rule, after phage isolation, phage-DNA sequencing is mandatory to exclude resistance, lysogeny or virulence factors encoded by the phage. High titers and low quantities of endotoxin in the solution are also requirements before phage administration, and several in vitro experiments, such as killing curves or studies of synergy with drugs, are also desirable. Taking all this into account, patients suffering from chronic infections are currently the most suitable to receive this therapy when these steps need to be performed with the new phages, as it is shown in the case reports reviewed here. Nevertheless, this scenario would change by only generating large collections of phages ready to be used with characterized phages or at least with their DNA sequenced since the time needed to test susceptibility to phages and antibiotics is the same. Although the number of phages to test would be higher than the number of drugs, it would be possible to use phages in acute infections. Importantly, another clinical consequence of the narrow host range of phages is that no empiric phage treatment should be recommended in infections. This therapy needs to isolate the bacterium causing the infection to be effective, indicating that the possibilities of success of universal cocktails would be scarce.

One relevant issue in phage therapy is bacterial resistance to bacteriophages. Bacterial pathogens can develop resistance to bacteriophages as they do against antibiotics. Cocktails of phages diminish bacterial recurrence, however, the most effective treatment against resistant bacteria is a combination of phages and antibiotics. Phage resistance produces bacterial changes that conversely reduce their fitness and/or resensitize bacteria to previous resistant antibiotics. More in vitro experiments of phage/antibiotic combinations are needed to understand the synergy between these therapies, there is a lack in this regard in recent studies. This combination is the most realistic alternative for human therapy, as it is shown in clinical case reports in which bacterial eradication was achieved only after administering a cocktail of phages combined with antibiotics (and no with phages without antibiotics).

Importantly, MOIs and timing of phage administration have been shown to be crucial for in vitro and in vivo experiments to reach bacterial clearance. Nonetheless, these two parameters cannot be optimized in human treatments because the internal concentration of bacteria in patients is unknown and the patient usually receives phage therapy after the failure of antibiotic therapy. These parameters only can serve for biotechnological industry and as a general guide to understand that high titers should be administered as soon as possible in patients.

A topic that remains under investigation is the behavior of the immune response against identical deliveries of phages. Some authors affirm that the phage concentration gets reduced in the human body after several administrations of the same phage due to the immune system, while others describe no adverse consequences of identical challenges. More in vivo studies and clinical trials are needed to clarify this point, including pharmacokinetic and pharmacodynamic studies using different ways of phage administration.

To conclude, we want just to mention a decisive factor in the clinical practice that is not discussed in this review: the regulatory framework, which has been reviewed recently [105]. In general, phages can be easily used as the last resource in a hopeless patient. Nevertheless, the strict regulation of phages as commercial products for humans hinders the development of companies interested in phage therapy and abates the number of clinical trials.

## Figures and Tables

**Table 1 antibiotics-10-00672-t001:** Summary of in vivo studies using phage therapy against *A. baumannii*.

Infection Model	Bacteria	Phage Therapy	Antibiotic Combination	Outcome	References
Ex-vivo human lung epithelial A549 cells	10^6^ CFU of XDR strains	Φkm18p at MOIs of 0.01, 0.1 and 1	No	Cell survival	[24]
HeLa cells	10^7^ CFU of AB1 strain in 100µL of DMEM	Abp (10^8^ PFU in 100 µL) after 2 h	No	Survival of treated cells was similar to the positive control group at 24 h	[25]
Ex-vivo human heat-inactivated plasma blood and *G. mellonella*	Clinical strain	vB_AbaP_AGC01	Phage alone and combined with gentamicin, ciprofloxacin and meropenem	Increased survival with antibiotic combination	[26]
*G. mellonella*	Ab177_GEIH-2000	Ab105-2phiΔCI	Alone, imipenem or meropenem	Increased survival with antibiotic combination after 72 h	[27]
*G. mellonella*	Carbapenem-resistant strains	WCHABP1 and WCHABP12	No	Survival increased from 20% to 75% in treated larvae	[28]
*G. mellonella* and murine model of bacteremia	10^5^ CFU of ESBL strains in larvae model and 6 × 10^7^ CFU in murine model	vB_AbaM_3054 and vB_AbaM_3090 via IP in mice 2 h post-infection	No	100% survival after 80 h post-infection in larvae model, and 100% survival in murine model after 7 d	[29]
*G. mellonella* and murine model of acute pneumoniae	Carbapenem-resistant strain	BΦ-R2096 at MOIs of 10 and 100 for larvae, and 0.1, 1 and 10 MOIs for mice	No	At 48 h, a MOI of 100 reached 50% of survival of larvae while a MOI of 10 obtained 10%. Only MOI of 1 exhibited 100% of survival in mice after 12 d	[30]
Rat wound model	5 × 10^8^ CFU/mL	vB-GEC_Ab-M-G7	No	Efficacy was achieved. Treated rats reduced symptoms and bacterial load by 5 log	[31]
Full-thickness dorsal infected wound model	MDR AB5075	Five-member cocktail	No	Neither increase of size nor necrosis was visualized in treated mice. Non-mature biofilm was present on treated mice	[32]
Wound model in uncontrolled diabetic rats	MDR strain	48 h post-infection	No	Treated mice reduced inflammation and no bacteria was isolated at day 8	[33]
Mouse model of sepsis	IP 10^6^ CFU of AB900 and A9844	ΦFG02 and ΦCO01	No	Reduction of bacterial loads of treated mice with isolation of resistant strains sensitive to antibiotics	[34]
Mouse model of pneumonia	10^8^ CFU of MDR strain with IN infection	Cocktail of PBAB08 and PBAB25 injected from day −1 to day +7 (10^9^ PFU)	No	100-fold reduction in lungs was obtained in treated mice respect to the control group. In addition, inflammatory response was studied after IN, IP and oral routes and only IP phages increased 20% IgE compared to controls	[35]
Mouse model of sepsis	5 × 10^7^ CFU of panresistant ABZY9	Immediate IP injection of Abp9 at MOI of 10	No	8 out of 12 treated mice survived	[36]
Mouse model of sepsis	10^9^ CFU/mL	IP inoculation of 10^9^ PFU vB_AbaP_PD-6A3 1 h post-infection	No	A survival rate of 60% was obtained compared to 0% of control group	[37]
Mouse model of sepsis	2 × 10^7^ CFU of AB9	10^8^ PFU of vB-AbaS-D0 and -D2 IP alone and combined	No	vB-AbaS-D2 and a mix of the 2 phages reached 90 and 100% of survival, respectively. vB-AbaS-D0 showed 50% of survival after 7 d	[38]
Mouse model of sepsis	AB3	Cocktail	No	100% of survival 6 weeks after infection	[39]
Mouse model of pneumonia	IN carbapenem-resistant strain	BΦ-C62	No	3 d post-treatment, no bacteria was found in lungs with a concomitant improvement of histological damage	[40]
Mouse model of sepsis	ESBL strain (2–3 × 10^8^ CFU/mouse BALB/c and 6 × 10^8^ CFU/mouse C57BL/6)	ϕkm18p at MOIs of 0.1, 1 and 10 after 10 min and 1 h from infection	No	100% of survival in mice treated after 10 min and 56% of survival in BALB/C and 46% in C57BL/6 treated after 1 h	[41]
Mouse model of pneumonia in neutropenic mice	Carbapenem-resistant strain	SH-Ab15519 via IN 1 h post-infection at MOIs of 0.1, 1 and 10 and 2 h post-infection at 10 MOI	No	90% of survival was obtained in all mice treated after 1 h. Mice treated 2 h post-infection showed 66.7% after 14 d	[42]
Mouse model of pneumonia in neutropenic mice	2 × 10^8^ CFU MDR strain via IN	IN vB_AbaM-IME-AB2 at MOI of 0.1, 1 and 10. A MOI of 10 was administered 1 h, 4 h and 24 post-infection	No	Only a MOI of 10 obtained 100% of survival, and only mice treated 1 h post-infection showed 100% of survival	[43]

Abbreviations: CFU, colony-forming units; ESBL, extended-spectrum beta-lactamase; IN, intranasal; IP, intraperitoneal; MDR, multidrug resistant; MOI, multiplicity of infection; PFU, plaque-forming units; XDR, extensively drug resistant.

**Table 2 antibiotics-10-00672-t002:** Summary of in vivo studies using phage therapy against *K. pneumoniae* from 2019 to date.

Infection Model	Bacteria	Phage Therapy	Antibiotic Combination	Outcome	References
Zebrafish	10^3^ CFU IM	10^8^ PFU/mL IM 2 h post-infection	Alone or with streptomycin	77% reduction of bacterial load with phage alone and 98% reduction combined with streptomycin	[49]
*G. mellonella*	ST258 KL106 and ST23 K1	vB_KpnP_KL106-ULIP47 and vB_KpnP_KL106-ULIP54	No	Survival was similar to control the group for both strains	[50]
*G. mellonella*	ESBL-producing strain	MOI of 1, 10, 100 and 1000 of KP1801	No	>93% of survival was found for all MOIs as prophylaxis. Lower therapeutic effect	[51]
Mouse model of sepsis	IP 5 × 10^7^ CFU MDR ST258 strain	IP Pharr and ΦKpNIH-2 at different MOIs and times	No	Survival depended more on time than on dose, with good results at 1 h post-infection	[52]
Mouse model of sepsis	4 × 10^7^ CFU IP	2 × 10^8^ PFU IP at −2, 0 and 2 h compared to infection.	No	After 7 d, 100% of survival was reached in treated mice at −2 and 0 h. Treatment of 2 h post-infection obtained 60% of survival	[53]
Mouse infection model	2 × 10^8^ CFU K24 carbapenem-resistant 533 strain	1.7 × 10^8^ PFU after 10 min or 1 h post-infection IP of vB_KpnS_Kp13	No	100% of survival in phage administration after 10 min. 12.5% of survival after 1 h	[54]
Mouse model of pneumonia	IN 10^9^ CFU	IN VTCCBPA43 (tolerant to 80 °C) 2 h post-infection	No	Treated mice experienced a reduction in bacterial load and less lesions 48 post-infection. Phages were detected 6 d after infection	[55]
Mouse model of sepsis	IP 10^8^ CFU	Cocktail of phages	No	1 and 2 MOIs produced 100% of survival, while 0.01 MOI was less effective. Cocktails obtained better results	[56]

Abbreviations: CFU, colony-forming units; ESBL, extended-spectrum beta-lactamase; IM, intramuscular; IN, intranasal; IP, intraperitoneal; MDR, multidrug resistant; MOI, multiplicity of infection; PFU, plaque-forming units.

**Table 3 antibiotics-10-00672-t003:** Summary of in vivo studies using phage therapy against *E. coli* from 2010 to date.

Infection Model	Bacteria	Phage Therapy	Outcome	References
*G. mellonella*	10^6^ CFU/mL of 31 strains	10^3^ PFU/mL myPSH1131	A single dose of phage was enough to reduce the bacterial load, but 3 doses were necessary to achieve survival	[65]
*G. mellonella*	20 µL with 10^8^ CFU/mL	20 µL of 10^4^ PFU/mL of ec311, doses every 6 h	3 doses were necessary to achieve 100% survival	[66]
Model of gut colonization in mice	EAEC O104:H4 55989Str strain	Cocktail of CLB_P1, CLB_P2 and CLB_P3 via oral by drinking water	Bacterial concentration got reduced after 24 h, but after phage withdraw, bacterial regrew	[67]
Model of gut colonization in rats	EAEC, EHEC, EIEC, EPEC ETEC, DAEC	Mix of 140 phages for 20 d via drinking water and oral injection or feeding with vegetable capsules	Growth of exogenous *E. coli* flora was suppressed	[68]
Mouse model of gut infection	10^6^ CFU of Entretoaggregative strain	4 × 10^8^ PFU of PDX	A reduction of the goal target bacteria was achieved in murine feces without dysbacteriosis, but not in human feces in vitro	[69]
Mouse model of sepsis	10^8^ CFU/mL	IP 5 × 10^9^ CFU/mL of a cocktail	100% of survival in treated mice after 100 h	[56]
Mouse model of sepsis	IV K1 IHE3034	10^7^ PFU of IK1	High protection. Phages were accumulated in spleen	[70]
Mouse model of muscular infection	IM 40–50 µL of 10^8^ CFU/mL of O18:K1:H7	10^6^ PFU of K1 dep and K1 ind	K1dep resolved 100% of infections and K1 idp resolved 30% of cases	[71]
Mouse model of pneumonia	IN 536 bioluminescent and VAP strain	536_P1 and 536_P7 with MOI of 0.3, 3 and 10	100% of survival in all treated mice	[72]
Rabbit ileal loop	0.5 mL of 10^8^ CFU/mL O157:H7	Cocktail with 0.5 mL of 10^6^ PFU/mL of PAH6 and P2BH2	Reduction of accumulation of liquid in loop and decreased bacteria load	[73]

Abbreviation: CFU, colony-forming units; MOI, multiplicity of infection; PFU, plaque-forming units; IM, intramuscular; IV, intravenous; VAP, ventilator associate pneumoniae.

**Table 4 antibiotics-10-00672-t004:** Summary of in vivo studies using phage therapy against *P. aeruginosa* from 2015 to date.

Infection Model	Bacteria	Phage Therapy	Antibiotic Combination	Outcome	References
*G. mellonella* and mouse	10^5^ CFU/mL YMC11/02/R656 strains and IN in mouse	Bφ-R656 and Bφ-R1836 at MOIs of 100, 10 and 1 (IN in mice)	No	Treatment with Bφ-R656 and Bφ-R1836 increased 50 and 60% of survival in larvae and 66 and 83% in mice, respectively	[85]
*G. mellonella* and mouse	10^9^ CFU/mL PAK-lumi in larvae and 10^7^ in mouse IN	Cocktail with PYO2, DEV, E215, E217, PAK_P1 and PAK_P4 phages (IN in mice)	No	In larvae: MOI of 8 increased survival from 17% to 49% and MOI of 25 increased to 63% after 20 h. Prophylaxis also was provided. In mice: 100% of survival with 0.05 and 1 MOIs	[86]
Mouse model of pneumoniae	2 × 10^6^ CFU/mL IN	PELP20 IN administration 24, 36, 48, 72, 144 and 156 h post-infection	No	Complete clearance in 100% of mice treated at 24, 36, 48 and 72 h, and 70% of clearance in mice treated at 144 and 156 h	[87]
Mouse model of pneumoniae	2.5 × 10^6^ CFU FADDI-PA001 intratracheal administration	2 × 10^7^ PFU/mg intratracheally aerosolized PEV20 phage	No	Bacterial burden reduction of treated mice, from 1.3 × 10^10^ CFU to 6 × 10^4^ CFU in lungs	[88]
Mouse model of pneumoniae	MDR strain	1 mg with 10^6^ PFU PEV20 aerosolized into the trachea 2 h post-infection	Ciprofloxacin (0.33 mg) intotracheal aerosolized 2 h post-infection	Combined treatment with antibiotic reduced bacterial load by 5.9 log_10_	[89]
Preventive mouse infection model	2.5 × 10^7^ CFU/mL IN	1.2 × 10^9^ PFU/mL IN cocktail 48 h prior infection	No	After 24 h of bacterial challenge, more than 70% of pre-treated mice cleared the infection and the other 30% harboured up to 20 CFU/mL	[90]
Zebrafish	30 CFU/embryo	5 × 10^9^ PFU/mL of a cocktail of four phages	Ciprofloxacin 100 µg/mL	CF embryos reduced lethality from 83% to 52% and antibiotic combination increased survival	[74]

Abbreviations: CF, cystic fibrosis; CFU, colony-forming units; IN, intranasal; MOI, multiplicity of infection; PFU, plaque-forming units.

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
