# Peer review of "Advances in Bacteriophage Therapy against Relevant MultiDrug-Resistant Pathogens"

_antibiotics, 2021, doi:10.3390/antibiotics10060672_

Round 1

Reviewer 1 Report

The review article titled ‘Advances in bacteriophage therapy against relevant multidrug resistant pathogens’ by Lavado et al. provide an overview of phage therapy against certain multidrug resistance bacteria. There are some minor English language edits required. My comments are as follows

  1. The argument made in line 60 is not valid. Antibiotics are widely used in regular laboratory procedures such as cell culture, plasmid preparations etc. Thus, it is likely to yield more results in pubmed search. The authors do not mention if they searched for titles only. I recommend removing this comparison from the review.
  2. My major concern is this review mostly enlists previous studies and describes the results but does not provide any synthesis from those results. The authors do not need to mention each and every result and study in detail. There should always be some input from the authors at the end of each section.
  3. The authors have tables and text about the same study which makes is redundant.
  4. The review is very wordy. The authors should make conscious efforts to reduce to the length of the review.

Author Response

REVIEWER 1:

  1. The argument made in line 60 is not valid. Antibiotics are widely used in regular laboratory procedures such as cell culture, plasmid preparations etc. Thus, it is likely to yield more results in Pubmed search. The authors do not mention if they searched for titles only. I recommend removing this comparison from the review.

RESPONSE: We agree with the reviewer and we have deleted this comparison.

  1. My major concern is this review mostly enlists previous studies and describes the results but does not provide any synthesis from those results. The authors do not need to mention each and every result and study in detail. There should always be some input from the authors at the end of each section.

RESPONSE: We appreciate this comment and we agree with the reviewer about the enlist of previous studies. In consequence, we have modified the structure and we have substituted all in vivo studies by Tables 1, 2, 3 and 4 in order to let the reader to have the summarized information but, more importantly, we have added inputs of in vivo and case reports studies for the four species.

  1. The authors have tables and text about the same study which makes redundant.

RESPONSE: We agree with the reviewer and we have passed Tables 1, 2 and 3 to the Supplementary material and now they are Tables S1, S2 and S3. (We would like to explain that the reviewer 2 liked our Tables and as a result we have chosen to put them as Supplentary material instead of deleting them).

  1. The review is very wordy. The authors should make conscious efforts to reduce to the length of the review.

RESPONSE: We thank this comment in order to improve the manuscript and to do so, we have shortened some sentences to avoid tedious sentences and also the review has been shortened in length after the substitution of the four in vivo written sections by four Tables.

Reviewer 2 Report

The authors provide a comprehensive and valuable review of recent experimental studies, case reports and clinical trials on the efficacy of phage therapy. The review is largely ready to be published and I have no major concerns. The manuscript would mostly just benefit from another round of proof reading – I have found a number of small errors throughout the manuscript, and I suspect that I missed several.

Below are the line by line comments:

Line 10-11: grammar? …of critical priority to develop sounds odd to me

Line 13-15: shorten sentence. Both the intro and discussion especially have several very long sentences that are hard to read.

Line 25: no THE

Line 46: split sentence in half

Line 51: split sentence in half

Line 53-55: since this is a review, perhaps a quick explanation of what virulent/lytic and temperate/lysogenic means would be helpful? Or a citation to a paper that does it.

Line 56: Citations?

Line 66: What does afresh mean? Grammar?

Line 77: Being a nosocomial pathogen, this organisms can cause…

Line 99: perhaps tables similar to 1-3 can be used to also sum up these reports. Same for 3.1, 4.1., 5.1. Similarly, it would perhaps be helpful if sample sizes were included in description of these studies – especially when things like 100% survival or so are mentioned.

Line 105: No comma after and needed

Line 111: sometimes units of time (hours, days) are written out, sometimes h and d are used.

Line 111: Write out PFU when first used – same for all other abbreviations.

Line 193: IP injection should be spelled out the first time it is used

Line 198: Lysins are never mentioned in the introduction, but perhaps should be.

Line 244: IN should be spelled out when first used. Line 370 spells out IM so it should be consistent.

Line 277: Compassionate use should be mentioned in the introduction

Line 561: Specific 140 bacteriophages – I don’t quite understand the wording here

Line 638: 6log10 - logarithms should be written uniformly throughout the paper

Line 743: There is no degree of homology, something is either a homolog or not. Perhaps similar is the better word?

Line 809: days not day

Line 876: For clinical trials, please also include the sample size

Line 891: Use “While” instead of despite

Line 896: “pretend” implies that that is not the true goal of the study.

Line 925: compareD

Line 995: COMMA although it has concluded

Line 974: this study or these studies?

Line 981: no posted results

Line 982: pretendes?

Line 990: for humans COMMA although

Line 991: which enables the targeting of desired bacteria

Line 994: which lengthens the development process

Line 994: this is a very long sentence

Tables: Several words are cut off and split between two lines, which looks strange. I’d make sure these are properly formatted before publication. Also several spelling/grammar errors.

Author Response

REVIEWER 2

Line 10-11: grammar? …of critical priority to develop sounds odd to me

RESPONSE: We thank this comment and we have substituted by “critically priority” in current line 11.

Line 13-15: shorten sentence. Both the intro and discussion especially have several very long sentences that are hard to read.

RESPONSE: We have shortened the sentence with a period in line14. We have also modified some sentences from the Introduction and Discussion sections to avoid long sentences.

Line 25: no THE

RSPONSE: Corrected.

Line 46: split sentence in half

RESPONSE: Modified (period in line 46).

Line 51: split sentence in half

RESPONSE: Modified (period in line 51).

Line 53-55: since this is a review, perhaps a quick explanation of what virulent/lytic and temperate/lysogenic means would be helpful? Or a citation to a paper that does it.

RESPONSE: We agree with the reviewer and we have added a reference in order to clarify the concept (line 54, reference number 7).

Line 56: Citations?

RESPONSE: We have added a reference (line 58, reference 9).

Line 66: What does afresh mean? Grammar?

RESPONSE: We have substitute “Afresh” by “Recently,”.

Line 77: Being a nosocomial pathogen, this organism can cause…

RESPONSE: Modified (line 84).

Line 99: perhaps tables similar to 1-3 can be used to also sum up these reports. Same for 3.1, 4.1., 5.1. Similarly, it would perhaps be helpful if sample sizes were included in description of these studies – especially when things like 100% survival or so are mentioned.

RESPONSE: We have added Tables 1, 2, 3 and 4 for in vivo studies and inputs of those sections.

Line 105: No comma after and needed

RESPONSE: Modified (line 113).

Line 111: sometimes units of time (hours, days) are written out, sometimes h and d are used.

RESPONSE: We agree with the reviewer and we have unified the criteria of writing h and d through the text and tables.

Line 111: Write out PFU when first used – same for all other abbreviations.

RESPONSE: We agree with the reviewer and we have written the complete name of abbreviations the first time it appears in the manuscript.

Line 193: IP injection should be spelled out the first time it is used

RESPONSE: Corrected.

Line 198: Lysins are never mentioned in the introduction, but perhaps should be.

RESPONSE: We have included a sentence about lysins in the Introduction section with one reference (lines 54-56, reference 8).

Line 244: IN should be spelled out when first used. Line 370 spells out IM so it should be consistent.

RESPONSE: Corrected.

Line 277: Compassionate use should be mentioned in the introduction

RESPONSE: Compassionate use has been included in the Introduction section with one reference (lines 71-78, reference 11).

Line 561: Specific 140 bacteriophages – I don’t quite understand the wording here

REFERENCE: This information has been passed to Table 3. The authors made a preventive cocktail mixing 140 phages.

Line 638: 6log10 - logarithms should be written uniformly throughout the paper

REFERENCE: Corrected.

Line 743: There is no degree of homology, something is either a homolog or not. Perhaps similar is the better word?

RESPONSE: We agree with the reviewer. Corrected (line 455).

Line 809: days not day

RESPONSE: Corrected.

Line 876: For clinical trials, please also include the sample size

RESPONSE: We have added the sample size of clinical trials (lines 596, 603, 607, 611, 665, 674, 681 and 691).

Line 891: Use “While” instead of despite

RESPONSE: Corrected (line 598)

Line 896: “pretend” implies that that is not the true goal of the study.

RESPONSE: We have substitute it by “aims to” (line 604).

Line 925: compared

RESPONSE: Corrected (line 633).

Line 955: COMMA although it has concluded

RESPONSE: Corrected (line 664).

Line 974: this study or these studies?

RESPONSE: Corrected, is “this study” (line 683).

Line 981: no posted results

RESPONSE: Corrected (line 690).

Line 982: pretendes?

RESPONSE: Corrected: “intends” has been included in line 691.

Line 990: for humans COMMA although

RESPONSE: Corrected (line 697).

Line 991: which enables the targeting of desired bacteria

RESPONSE: Corrected (line 699).

Line 994: which lengthens the development process

RESPONSE: Corrected (lines 701-702).

Line 994: this is a very long sentence

RESPONSE: The sentence has been split in two (lines 702-704).

Tables: Several words are cut off and split between two lines, which looks strange. I’d make sure these are properly formatted before publication. Also several spelling / grammar errors.

RESPONSE: We agree with the reviewer. Tables have been reviewed and modified.

Round 2

Reviewer 1 Report

The authors have satisfactorily answered the queries.